# High-performance p-channel transistors with transparent Zn doped-CuI

Ao Liu[1,5], Huihui Zhu[1,5], Won-Tae Park [2], Se-Jun Kim[3], Hyungjun Kim[3], Myung-Gil Kim [4✉] & Yong-Young Noh[1✉]

'Ideal' transparent $p$-type semiconductors are required for the integration of high-performance thin-film transistors (TFTs) and circuits. Although CuI has recently attracted attention owing to its excellent opto-electrical properties, solution processability, and low-temperature synthesis, the uncontrolled copper vacancy generation and subsequent excessive hole doping hinder its use as a semiconductor material in TFT devices. In this study, we propose a doping approach through soft chemical solution process and transparent $p$-type Zn-doped CuI semiconductor for high-performance TFTs and circuits. The optimised TFTs annealed at 80 °C exhibit a high hole mobility of over 5 cm$^2$ V$^{-1}$ s$^{-1}$ and high on/off current ratio of ~10$^7$ with good operational stability and reproducibility. The CuI:Zn semiconductors show intrinsic advantages for next-generation TFT applications and wider applications in optoelectronics and energy conversion/storage devices. This study paves the way for the realisation of transparent, flexible, and large-area integrated circuits combined with $n$-type metal-oxide semiconductor.

[1] Department of Chemical Engineering, Pohang University of Science and Technology, Pohang, Gyeongbuk 37673, Republic of Korea. [2] Department of Electrical and Computer Engineering, University of Waterloo, 200 University Avenue West, Waterloo, ON N2L 3G1, Canada. [3] Department of Chemistry, Korea Advanced Institute of Science and Technology, Daehak-ro 291, Yuseong-gu, Daejeon 34141, South Korea. [4] School of Advanced Materials Science and Engineering, Sungkyunkwan University, Suwon 16419, Republic of Korea. [5] These authors contributed equally: Ao Liu, Huihui Zhu. ✉email: myunggil@skku.edu; yynoh@postech.ac.kr

Since the commercialisation of the *n*-type metal-oxide semiconductor *a*-InGaZnO (*a*-IGZO) for thin-film transistors (TFTs) in flat panel displays in 2011, transparent *p*-type counterparts have attracted increasing interest for high-performance complementary logic circuits and next-generation 'invisible' active-matrix organic light-emitting diode displays[1–4]. However, considering the industrial requirements of high mobility and optical transmittance, despite the extensive studies, the reported *p*-type metal-oxide semiconductors still exhibit insufficient performance[5]. The poor electrical properties originate from their inherent drawbacks of localised hole transport path (i.e., oxygen 2*p* orbitals) in the valence band maximum (VBM) and strong self-compensation during the doping[6–8]. Although the concept of chemical modulation of the valence band was effective for fabricating new transparent *p*-type materials, such as $CuMO_2$ delafossites (M = Al, In, Ga, etc.) and LnCuOCh oxychalcogenides (Ln = lanthanide, Ch = chalcogen)[2,9], the low hole mobilities or high carrier concentrations and high deposition temperatures (>700 °C) make them unsuitable for transistor applications. Therefore, the search for transparent *p*-type semiconductors beyond oxides with excellent hole-transport properties and low-temperature synthesis techniques has attracted considerable interest in recent years.

To overcome the poor band dispersion of VBM with small oxide anions, alternative inorganic materials with large and easily polarisable anions have been investigated as novel *p*-type transparent semiconductors[10]. Among them, copper(I) iodide (CuI) is regarded the most promising candidate owing to its high intrinsic Hall mobility over $40 \, cm^2 \, V^{-1} \, s^{-1}$, high optical transparency with a wide bandgap ($E_g$) of ~3 eV, and high doping capacity with a significant *p*-type conductivity[11,12]. CuI is consisted of abundant elements and can be synthesised at low temperatures, which enables various applications on flexible plastic substrates[13,14]. CuI also shows diverse application potentials in thermoelectric devices, heterojunction diodes, and as a transparent conductor and hole-transport layer in photovoltaic devices[12–19]. Initial studies on its applications as a semiconductor channel layer in TFTs have been reported recently[20–22]. However, considering the intrinsically facile metal vacancy generation and subsequent excessive hole ($n_h > 10^{19} \, cm^{-3}$), the resultant TFTs exhibited low hole mobilities and poor current modulation capabilities with low on/off current ratios ($I_{on}/I_{off}$) of ~$10^2$. To reduce the number of holes for the realisation of a high-performance CuI-based TFT, a feasible approach is to create donor-like iodine vacancies by thermally decomposing the lattice iodine, as demonstrated in our recent study[20]. However, the significant iodine vacancy generation and undesired grain aggregation limited the further improvement in TFT performance and were fatal to the device operational stability.

In this study, a solution-based doping approach was proposed to screen suitable metal cations as hole suppressors for CuI with the combination of theoretical and experimental routes. Considering the ionic radius and local geometry to those of $Cu^+$, $Zn^{2+}$ was demonstrated as the champion *n*-type dopant, enabling high-performance p-channel TFT fabrication at 80 °C with great reproducibility over large area. The further integration of logic inverter with a *a*-IGZO TFT exhibits excellent performance with a high gain of 56.

## Results

### Material design of CuI with a hole-suppressor dopant. CuI has a zincblende structure below 643 K ($\gamma$-CuI) with intrinsic copper vacancies as hole producers at room temperature[11]. It has a highly dispersed valence band with an effective mass of $0.3m_0$ for light holes, comparable to those of high-mobility *n*-type oxides as complementary circuit components (Supplementary Table 1).

However, the facile copper vacancy formation by the thermal stress during the fabrication or off-stoichiometry induces a significant Fermi level shift to near the VBM with a high hole concentration of $10^{19} \, cm^{-3}$ for the as-deposited film[11,12,23]. To address this issue, the substitutional doping of the semiconductor with different-valence atoms can be utilised as a standard process to control the carrier concentration. The lattice structure of CuI is shown in Fig. 1a. Its VBM is characterised as a hybridisation of Cu 3*d* and I 5*p* states, while the conduction band minimum (CBM) is characterised as Cu 4*s* states (Supplementary Fig. 1a). The Cu vacancies in the CuI lattice down-shift the Fermi level, and thus create shallow defects adjacent to the VBM (upper panel of Fig. 1b). Considering the sizes and local geometries of the doped metal cations (Fig. 1c), $Zn^{2+}$ could be an ideal dopant for CuI as a hole suppressor. Upon the $Zn^{2+}$ doping, the defects were removed and the Fermi level was recovered to the forbidden region. Notably, this treatment produced intermediate states in the gap region as poor electron donors far from the CBM (~0.5 eV). These mid-states are characterised mostly as a hybridisation of Zn 4*s* and I 5*p* orbitals. Similar trends were observed for dopings of $Pb^{2+}$ and $Bi^{3+}$ at $Cu^+$ sites (Supplementary Fig. 1b, c), except that the mid-states of the $Bi^{3+}$-doped system were considerably closer to the Fermi level, far away from the CBM. This is attributed to the higher electropositivity of Bi than those of Zn and Pb, which favours the formation of the 3+ state instead of the 2+ state, and thus suppresses the creation of holes.

Consequently, our density functional theory (DFT) results suggest the control of the Fermi level location by properly choosing the dopant (among $Zn^{2+}$, $Pb^{2+}$, and $Bi^{3+}$). However, the introduction of larger ions with improper coordination, such as $Pb^{2+}$ and $Bi^{3+}$, induces a lattice distortion near the dopant (Supplementary Fig. 2). Therefore, stable dopings of $Pb^{2+}$ and $Bi^{3+}$ into CuI might be challenging to achieve by typical synthesis methods.

### Characterisations of doped CuI films. Various film characterisations were carried out to evaluate the doping effects of different metal cations (e.g., $Zn^{2+}$, $Ni^{2+}$, $Pb^{2+}$, $Bi^{3+}$, $Ga^{3+}$, and $Sn^{4+}$) on the film formation and properties. We focus on the $Zn^{2+}$-doped CuI system and the results from other dopants are presented in Supplementary Information. CuI:Zn precursor solutions with different $Zn^{2+}$ contents were prepared by mixing CuI and $ZnI_2$ acetonitrile solutions, followed by film coating and low-temperature annealing at 80 °C (see "Methods" section), which yielded films with negligible impurity contents (Supplementary Fig. 3). Figure 2a shows X-ray diffraction (XRD) patterns of CuI thin films doped with different cations at a concentration of 5 mol %. The preferential (111) orientation ($2\theta = 25.5°$) is assigned to the $\gamma$-phase CuI with the zincblende structure[23]. Owing to the relatively large radii of $Bi^{3+}$ and $Pb^{2+}$ and thus low doping efficiencies in CuI matrix, the separated $BiI_3$ and $PbI_2$ phases were detectable even with 2 mol% adding (Supplementary Fig. 4). This is consistent with the DFT calculation presented in Supplementary Fig. 2. The *p*-type conductivity of CuI mainly originates from Cu vacancies, which also affect the crystallisation[20]. When the small amount of $ZnI_2$ was doped into CuI, the $Zn^{2+}$ ions could fill or compensate Cu vacancies, which led to the improved film crystallinity (Fig. 2b). The Cu vacancy filling also slightly expanded CuI lattice, shifting the diffraction peak toward lower angle[24]. By contrast, due to the similar ionic radii of $Zn^{2+}$ (74 pm) and $Cu^+$ (77 pm), the $Zn^{2+}$ substitution on Cu sites kept the original lattice structure with negligible peak shift. Higher $Zn^{2+}$ doping amounts (≥10 mol%) formed a new phase of $Cu_2ZnI_4$.

The X-ray photoelectron spectroscopy (XPS) results in Fig. 2c show the linearly increasing Zn 2*p* peak intensity with the $Zn^{2+}$

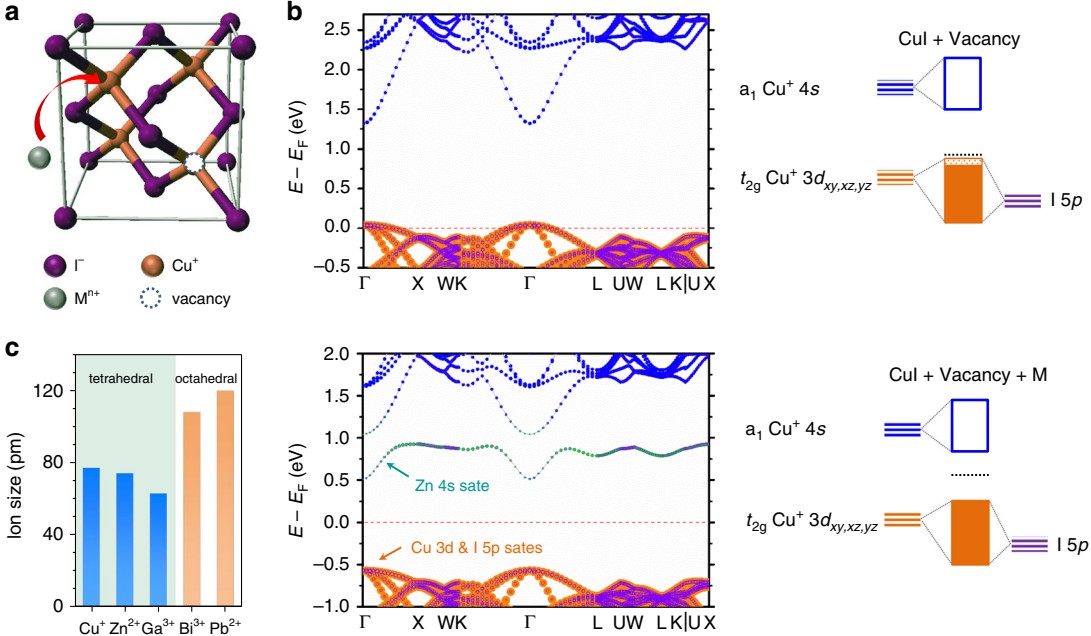

**Fig. 1 DFT calculation. a** Unit-cell of CuI and schematic of a Cu vacancy and doping at a $Cu^+$ site. **b** (Upper panel) Projected band structure of CuI with a Cu vacancy content of 3%; the Cu $3d$ (orange), Cu $4s$ (blue), and I $5p$ (purple) states are shown. The corresponding schematic showing the orbital characteristics of the VBM and CBM is also presented. (Lower panel) Projected band structure of CuI with a Cu vacancy content of 3% and $Zn^{2+}$ substitutional doping; the Cu $3d$ (orange), Cu $4s$ (blue), I $5p$ (purple), and Zn $4s$ (green) states are shown. The corresponding schematic showing the orbital characteristics of the VBM and CBM is also presented. **c** Radii and typical coordination of candidate dopant cations.

doping, which confirms the increased dopant concentration. The optical transmittance spectra in Supplementary Fig. 5 reveal that all CuI:Zn thin films are fully transparent in the visible region with wide bandgaps ($E_g$) of ~3 eV (Fig. 2d), which are desired for transparent electronics. The slightly increased $E_g$ with 5-mol% $Zn^{2+}$ doping could be attributed to the passivation of copper vacancies by $Zn^{2+}$, which reduced the density of states in the band structure. This property is unique because the considerably declined optical transmittance was noticed after adding other dopants (Supplementary Fig. 6). The atomic force microscopy (AFM) images in Fig. 2e show the nonuniform and agglomerated surface morphology of pristine CuI with a root mean square (RMS) roughness of 1.22 nm, which could be attributed to the high volatility of acetonitrile solvent and rapid crystallisation tendency of CuI film at 80 °C. The addition of a suitable amount of $Zn^{2+}$ effectively retarded the rapid crystallisation with a uniform grain distribution. The RMS roughness slightly increased to 1.43 and 1.60 nm for the 5- and 10-mol%-Zn-doped CuI thin films, which was mainly related to the enhanced film crystallinity. However, the further increase in $Zn^{2+}$ doping ratio to 15 mol% led to a rough surface (RMS roughness of 5.37 nm), mainly owing to the formation of segregation phase (Supplementary Fig. 7).

The microstructures of the CuI:Zn thin films were further analysed by transmission electron microscopy (TEM), as shown in Fig. 2f and Supplementary Fig. 8. The high-resolution TEM image shows a lattice spacing of 0.348 nm for pristine CuI, which corresponds to the (111) crystalline plane of γ-phase CuI. This preferential growth orientation was verified by an in situ fast Fourier transform (FFT) pattern of the selected area. When the $Zn^{2+}$ doping content was lower than 10 mol%, negligible variations in structure and interplanar spacing were observed and no diffraction pattern of the $ZnI_2$ phase could be detected. The $Zn^{2+}$ was uniformly distributed in the composite films, according to the energy-dispersive X-ray spectroscopy (EDS) mapping and Secondary-ion mass spectrometry (SIMS) analysis (Supplementary Figs. 9 and 10). However, upon the doping of

15-mol% $Zn^{2+}$, the continuous film changed to the combination of separated particles, which is consistent with XRD and AFM analyses.

The energy band variation of CuI thin film as a function of $Zn^{2+}$ doping was evaluated using ultraviolet photoemission spectroscopy (UPS). As shown in Fig. 2g, the secondary electron cut-off edge shifted towards higher energies upon the $Zn^{2+}$ addition, which indicated a Fermi energy level shift towards the conduction band edge (i.e., $n$-doping)[25]. The $Zn^{2+}$ occupation on Cu vacancies ($V_{Cu}$ has one negative charge) and the $Zn^{2+}$ substitution at $Cu^+$ sites can generate extra electrons and thus reduce hole concentration in CuI films. The VBM calculated by using the valence region was positioned at ~5.25 eV, well aligned with the work function of gold electrodes ($\Phi_{Au} = 5.1$ eV) for Ohmic contact. The corresponding energy band diagrams of different Zn-doped CuI samples are presented in Fig. 2h.

**Electrical characterisations of doped CuI TFTs and inverters.** To investigate the electrical properties of CuI thin films doped with different cations, a series of bottom-gate, top-contact TFTs were fabricated on $SiO_2/p^+$-Si wafers with thermally evaporated Au source/drain electrodes. The optimised channel thickness is ~9 nm for the achievement of both high mobility and $I_{on}/I_{off}$ (Supplementary Fig. 11). The channel layers were patterned through probe scratching to guarantee low gate leakage currents and reliable parameter extraction (Supplementary Fig. 12). The corresponding electrical characteristics are summarised in Fig. 3, Supplementary Fig. 13, and Table 2. Typical $p$-channel transistor behaviours were observed for all devices except for the $Sn^{4+}$-doped devices (inactive). The optimal results for each dopant in Fig. 3a show that $Zn^{2+}$ is the best dopant for CuI. The device performance as a function of $Zn^{2+}$-doping content are summarised in Fig. 3b and Supplementary Fig. 14. A low current modulation with a high turn-on voltage ($V_{on}$) of ~40 V was observed for the 2.5-mol% $Zn^{2+}$-doped CuI TFT, which reflected

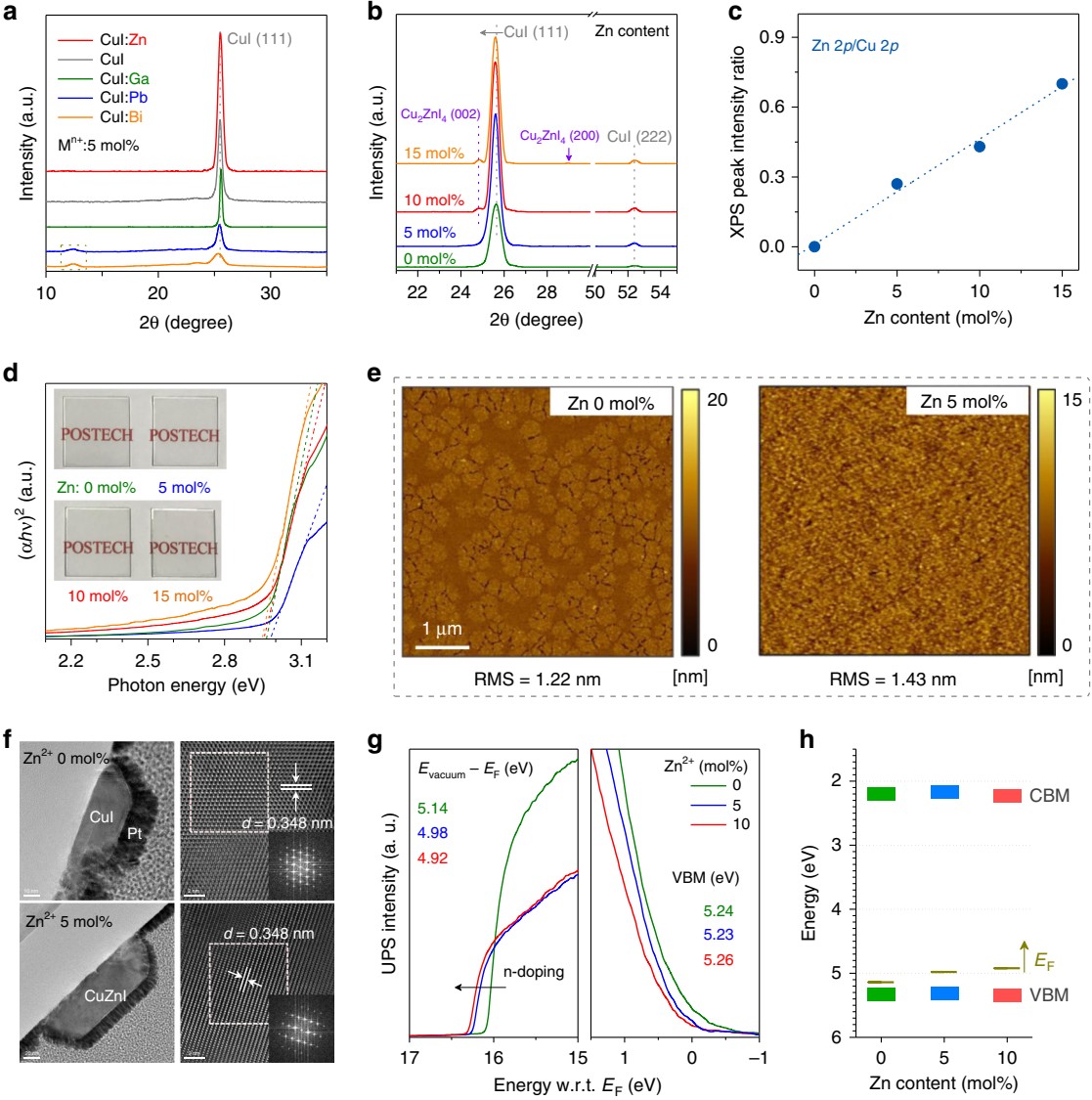

**Fig. 2 Characterisations of the doped CuI thin films. a** XRD patterns of the CuI thin films doped with different cations. **b–d** XRD patterns, XPS peak intensity ratios of Zn 2$p$ to Cu 2$p$, and Tauc plots of the Zn-doped CuI films with Zn doping contents of 0–15 mol%. **e, f** AFM and TEM images (FFT pattern) of the bare and 5-mol%-Zn-doped CuI films (scale bar in HRTEM images is 2 nm). **g, h** UPS spectra of the secondary-electron onset and valence band regions and electronic energy level diagrams for the CuI:Zn thin films with different Zn$^{2+}$ doping contents (0, 5, 10 mol%).

still high hole concentration in the channel layer. The further Zn doping reduced the current level and mobility in slope and shifted the threshold voltage ($V_{TH}$) in the negative direction (Fig. 3c). These trends indicate a decrease in channel conductivity with the Zn$^{2+}$ doping, which is consistent with the UPS and DFT calculation results. The film Hall-effect measurement results also showed the declined hole concentration and mobility with Zn$^{2+}$ doping (Supplementary Fig. 15). Overall, the TFT doped with 5-mol% Zn$^{2+}$ (CuI:Zn$_{5mol\%}$) exhibited well-compromised electrical performance, including a high on-state current of 1 mA, high saturation mobility ($\mu_{sat}$) of 5.3 ± 0.3 cm$^2$ V$^{-1}$ s$^{-1}$, a high $I_{on}/I_{off}$ of ~10$^7$, and $V_{TH}$ of 17.5 ± 2.0 V, respectively. Both mobility and $I_{on}/I_{off}$ are record-high compared with those of previously reported solution-processed $p$-channel metal-oxide/(pseudo) halide TFTs (mobility <1 cm$^2$ V$^{-1}$ s$^{-1}$ and $I_{on}/I_{off}$ ≤ 10$^4$, details in Supplementary Table 3 and Fig. 16).

Among the CuI TFTs with other dopants (Pb$^{2+}$, Ni$^{2+}$, Ga$^{3+}$, Bi$^{3+}$, Sn$^{4+}$, etc.), the 5-mol% Bi$^{3+}$ addition achieved the

optimised performances, including an effective current modulation, a $\mu_{sat}$ of 0.45 cm$^2$ V$^{-1}$ s$^{-1}$, and an $I_{on}/I_{off}$ of ~10$^4$. The inferior electrical performance compared with CuI:Zn TFTs can be attributed to the lower doping efficiency of Bi$^{3+}$ owing to its considerably larger radius (108 pm) than that of Cu$^+$ (77 pm) and improper local coordination preference (octahedral vs. tetrahedral). The scanty Bi$^{3+}$ substitution at Cu$^+$ sites could also disrupt the locally ordered CuI framework and create undesired defects, which impeded the hole transport. In addition, unusual $V_{TH}$ 'kink' and $\mu_{sat}$ disappearance were observed for the CuI:Bi$_{5mol\%}$ TFT (Supplementary Fig. 17), which could be attributed to the voltage-induced migration of charged defects or ionic species originated from the low-crystallinity CuI and segregated BiI$_3$. The accumulation of dissociative ions near the electrodes could screen the applied electric field, which reduced the field-modulation capability. The ionic migration in the Bi-incorporated CuI was confirmed by scanning transfer curves at lower speeds, which led to an increased current level (Supplementary Fig. 18). Meanwhile,

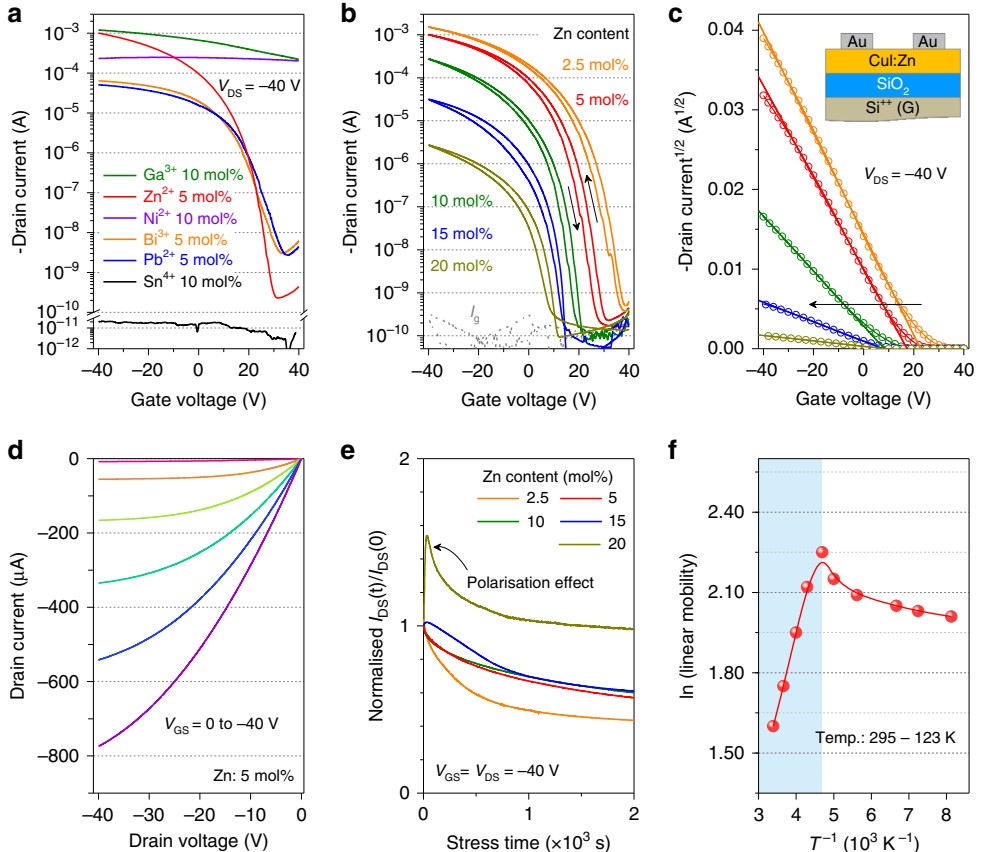

**Fig. 3 Electrical performance of doped CuI TFTs. a** Optimised transfer characteristics of the CuI TFTs doped with different cations ($Ga^{3+}$, $Zn^{2+}$, $Ni^{2+}$, $Bi^{3+}$, $Pb^{2+}$, and $Sn^{4+}$). **b, c** Transfer characteristics and $I_{DS}^{1/2}$ curves of CuI:Zn/SiO$_2$ TFTs with different $Zn^{2+}$-doping contents ($V_{DS} = -40$ V). **d** Output curves of optimised CuI:Zn$_{5mol\%}$/SiO$_2$ TFT. **e** Negative-bias-stress results of the CuI:Zn/SiO$_2$ TFTs as a function of $Zn^{2+}$ doping content. **f** Linear mobility variation of the CuI:Zn$_{5mol\%}$/SiO$_2$ TFT as a function of the temperature (295–123 K).

the $\mu_{sat}$ increased and $V_{TH}$ shifted positively with the appearance of 'kink' behaviour. In contrast, the CuI:Zn TFT exhibited a stable operation with a high degree of consistency independent on the scanning speed, indicating that CuI:Zn is electrically reliable for use in transistors.

To investigate the device stabilities with Zn-doped CuI semiconductors, the operational stability under negative-bias-stress test was first carried out (Fig. 3e). The $Zn^{2+}$ doping could effectively inhibit the drain current ($I_{DS}$) reduction under long-term fixed voltage application ($V_{GS} = V_{DS} = -40$ V), mainly owing to the passivation of traps, such as copper vacancies. When the $Zn^{2+}$ doping contents were lower than 10 mol%, the $I_{DS}$ decreased in slope under the bias stress, which was commonly observed owing to the charge trapping in the channel layer and at the channel/dielectric interface[26–28]. The interface trap densities ($D_{it}$, $D_{it} = \left[\frac{SS \times \log e}{kT/q} - 1\right]\frac{C_i}{q}$), where $k$ is the Boltzmann's constant, $T$ is the absolute temperature, and SS is the subthreshold swing, were calculated to be $1 \times 10^{13}$, $7.1 \times 10^{12}$, $6.7 \times 10^{12}$, $7.2 \times 10^{12}$, and $9.8 \times 10^{12}$ cm$^{-2}$ for 2.5-, 5-, 10-, 15-, and 20 mol% $Zn^{2+}$-doped CuI TFTs, respectively. Notably, when the $Zn^{2+}$ content was higher than 15 mol%, $I_{DS}$ initially increased, and then decreased. The anomalous $I_{DS}$ increase could be speculated by the stress-induced ionic conduction by the slow ion migration. Similar to the current variation trend, the $V_{TH}$ shifted negatively under bias stress without obvious subthreshold swing variation (Supplementary Fig. 19). This indicated the defect state creation was negligible, and charge trapping was the dominant instability mechanism. In addition, benefiting from the large $E_g$ of CuI:Zn

channel layers, the devices exhibited stable long-term operational stability under the visible light irradiation (Supplementary Fig. 20). For the CuI:Bi TFTs, owing to the low doping efficiency of $Bi^{3+}$ in CuI matrix, the abnormal $I_{DS}$ increase were observed in relatively long term under bias-stress tests (Supplementary Fig. 21). The air stability investigation indicated the impressionable feature of CuI:Zn semiconductor in air and the adoption of hydrophobic CYTOP passivation layer could improve the ambient durability (Supplementary Fig. 22).

The temperature-dependent measurement was then carried out to evaluate the charge transport in the CuI:Zn TFT (Fig. 3f and Supplementary Fig. 23). The mobility initially increased with the decrease in temperature from 295 to 213 K, which corresponds to the typical band-like transport behaviour[29–31]. Such conduction was commonly observed in high-mobility semiconductors[32–34]. The band-like transport could be attributed to the highly dispersed valence band and high degree of order in the channel layer. The subsequent mobility reduction in the lower-temperature range implies that the hole transport turned to a thermally activated transport, which could be attributed to shallow traps within grain boundaries or at the semiconductor/dielectric interface[11]. The thermally activated hole transport was also demonstrated in other inorganic Cu-based $p$-type semiconductors (e.g., CuSCN and Cu$_x$O) and was the main factor for the low mobilities[2,35,36].

To study the scalability and uniformity of CuI:Zn TFTs, we fabricated wafer-scale TFT array (>600 TFTs) on a 4-inch Si/SiO$_2$ (100 nm) substrate. The photograph in Fig. 4a shows uniform film coverage over the entire substrate. 96 devices were selected

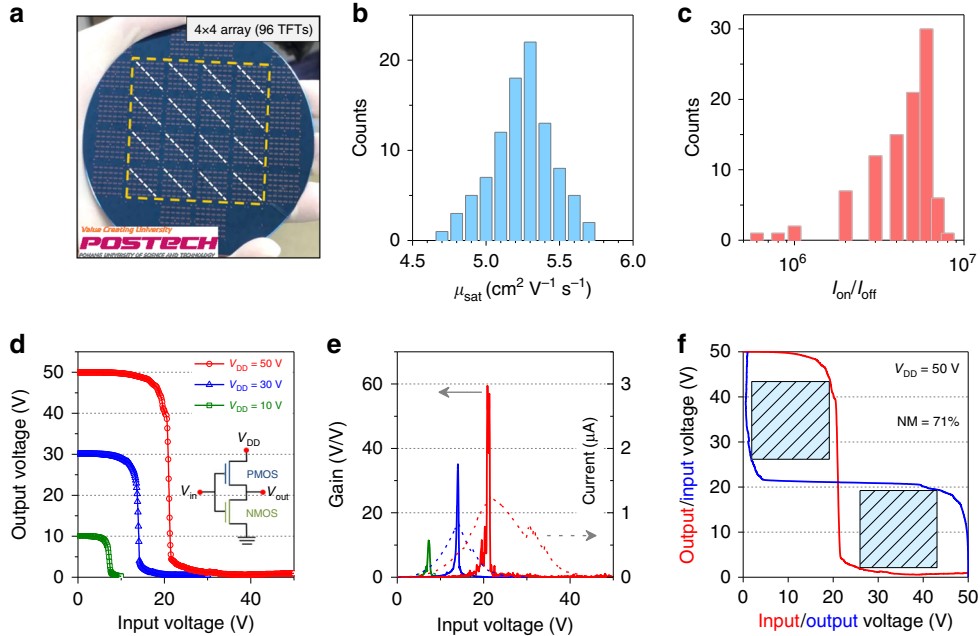

**Fig. 4 Wafer-scale uniformity and the CMOS inverter integration with n-channel IGZO TFTs. a** Photograph of CuI:Zn$_{5mol\%}$ TFT array on a 4-inch Si/SiO$_2$ (100 nm) wafer substrate (dot line means measurement area). **b, c** Statistical results of $\mu_{sat}$ and $I_{on}/I_{off}$ obtained from 96 TFTs across the array. **d–f** Voltage transfer, gain, current characteristics, and noise margin (NM) extraction of the complementary inverter based on *n*-type IGZO/SiO$_2$ and *p*-type CuI:Zn/SiO$_2$ TFTs.

and measured regularly along the dash lines and the corresponding transfer characteristics were summarized in Supplementary Fig. 24. The statistical distributions of the $\mu_{sat}$ and $I_{on}/I_{off}$ values are displayed in Fig. 4b, c. The data show high device yield with $\mu_{sat}$ of $5.3 \pm 0.5$ cm$^2$ V$^{-1}$ s$^{-1}$ and $I_{on}/I_{off}$ of $10^6$–$10^7$. The minor performance deviation is reasonable and can be related to the slight nonuniform thickness distribution in center and margin areas (<1 nm in this work) using spin-coating method. Finally, a proof-of-concept complementary inverter was assembled by cables to analyse the potentials of CuI:Zn TFTs for the realisation of high-performance logic circuits with *n*-channel *a*-IGZO devices (Supplementary Fig. 25). The voltage transfer characteristics, shown in Fig. 4d, exhibited full rail-to-rail swings and rapid voltage transitions with a high peak gain of 56 at a supply voltage ($V_{DD}$) of 50 V (Fig. 4e). The static currents ($V_{in} = 0$ V or $V_{in} = V_{DD}$) in the inverter were lower than 10 nA, which indicated that the static power consumption was smaller than 0.25 μW per logic gate (85 and 10 nW at 30 and 10 V $V_{DD}$). The noise margin, estimated by using the maximum equal criterion method[37], was higher than 70% of the ideal value ($V_{DD}$/2), which is sufficient for most static logic applications (Fig. 4f and Supplementary Fig. 26)[38–40]. The high gain, wide logic swing window, low power consumption, and excellent noise margin show the application potentials of the CuI:Zn *p*-type semiconductor for transparent circuits. Higher CMOS inverter performance is expected by optimising the circuit geometry and CuI:Zn TFT performance.

## Discussions

The above results demonstrate a new transparent inorganic *p*-type semiconductor of CuI:Zn, which enables the fabrication of high performance transistors. The thin films can be easily deposited using a one-step spin-coating process at plastic-compatible temperatures. Over the past 20 years, great efforts have been focused on *p*-type metal oxides, but we still have not realized the potential for practical applications because of too poor electrical performance, rigorous deposition process, and a lack of universality for easy repetition. Therefore, searching for

new inorganic transparent *p*-type candidates as oxide replacers should be considered. Herein, despite the initial attempt on CuI:Zn, its high transistor performance and easy processing capability exhibited great superiority when compared with previous reports. This work provides us an up-and-coming *p*-type semiconductor, which shows great compatibility with IGZO technology in the field of transparent electronics. However, the high-mobility devices were operated in depletion mode, which indicated a still high hole concentration in the channel layers. We believe that further improvements in the thin film quality (e.g., crystallinity and surface smoothness), interface modification, and device engineering will enhance the device performance and stabilities.

## Conclusions

We have proposed a new transparent inorganic *p*-type semiconductor (Zn-doped CuI) by spin coating at 80 °C. The band-like charge transport and stable doping enabled the fabrication of high-performance TFTs and inverter circuits with excellent reproducibility. The results demonstrate the promising application potential of CuI:Zn as active semiconducting components via simple process for large-area, low-cost, and transparent flexible electronics. The high doping capacity enabled the adjustment of *p*-type conductivity over a wide range, which is beneficial for the applications in printable optoelectronics and energy conversion/storage.

## Methods

**Preparation of the precursor solution**. All chemical reagents were purchased from Sigma-Aldrich and used as received without further purification. The CuI precursor solution (6 mg ml$^{-1}$) was prepared by dissolving a CuI powder into acetonitrile. Different doping concentrations were achieved by blending the CuI/acetonitrile solution with a dopant (ZnI$_2$, GaI$_3$, or NiI$_2$)/acetonitrile solution. The volume ratio of CuI/acetonitrile to dopant/acetonitrile was 10:1. For the optimized Zn-doped component, the ZnI$_2$ precursor concentration is 10 mg ml$^{-1}$. For the dopants of BiI$_3$, PbI$_2$, and SnI$_4$, *N,N*-dimethylformamide was used as a solvent owing to their poor solubility in acetonitrile. The solution mixing was carried out in a glove box (N$_2$) under stirring for 20 min before film casting.

**TFT fabrication**. A bottom-gate top-contact device structure was used in the TFT fabrication. Heavily doped Si substrates with 100-nm thermally grown $SiO_2$ gate dielectrics were used as gate electrode and dielectric layers. Mixed precursor solutions were filtered through a 0.2-μm syringe filter, and then spun on the plasma-treated hydrophilic $SiO_2$/Si substrates at 6000 rpm for 45 s. The samples were then annealed at 80 °C for 10 min with the thickness of ~9 nm. Au source and drain electrodes (40 nm) were deposited on the channel layers by thermal evaporation. The channel length and width of the shadow mask were 150 and 1000 μm, respectively. The CuI:Zn film coating should be carried out in inert glove box or air condition with low humidity (<20%). The fabrication procedures for the solution-processed IGZO can be found in our previous report[41].

**Thin-film characterisations**. The structures and compositions of the different CuI:X films were analysed by XRD (Rigaku, Ultima V) and XPS (PHI 5000 VersaProbe II). The film surface morphologies were analysed by tapping-mode AFM (Nanoscope V Multimode 8, Bruker). TEM (JEM 2100F) samples were deposited using focused ion beam (FIB). Hall measurement system (HMS-3000) was utilised to characterize the electrical property of CuI:Zn thin films. The optical transmittances of the thin films on quartz glasses were investigated by a UV–visible spectrophotometer (JASCO V-770). The optical band gap ($E_g$) was calculated by using the Tauc plot[42]

$$\alpha = \frac{1}{t} \ln \left[ \frac{(1-R)^2}{T} \right], \tag{1}$$

where $\alpha$ is the absorption coefficient

$$(\alpha h\nu)^2 = A\left( h\nu - E_g \right), \tag{2}$$

where $t$ is the sample thickness, $T$ is the transmittance, and $R$ is the reflectance, which could be neglected ($R \ll 1$).

**TFT electrical measurements**. Transistor measurements were performed by using a Keithley 4200 semiconductor parameter analyser in nitrogen glove box. The saturation TFT mobility was calculated as[7]

$$\mu_{sat} = \frac{2L}{WC_i} \left( \frac{\partial \sqrt{I_{DS}}}{\partial V_{GS}} \right)^2, \tag{3}$$

where $L$, $W$, and $C_i$ are the channel length and width and dielectric areal capacitance, respectively. The activation energy ($E_a$) in the low-temperature analysis was computed by fitting the data by using $\mu_{lin} = \mu_0 \exp(-E_a/k_B T)$, where $k_B$ is the Boltzmann constant.

**Computational methods**. The first-principle DFT calculations were performed by using Vienna ab-initio simulation package (VASP)[43] with the Perdew–Burke–Ernzerhof exchange–correlation functional[44]. The core electrons were described by using the projected-augmented-wave method[45]. The plane-wave energy cut-off for the valence electrons was set to 400 eV. The $d$-electrons of Zn, Pb, and Bi were included as valence electrons. The reciprocal space was sampled by using a $\Gamma$-centred $2 \times 2 \times 2$ mesh. We optimised the $2 \times 2 \times 2$ supercell of $\gamma$-CuI and removed one Cu atom to create a Cu vacancy. To model the doping system, Zn, Pb, or Bi was inserted to a position far from the vacant site by replacing one Cu atom.

## Data availability

All data related to this paper can be requested from the corresponding author upon reasonable request.

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

## Acknowledgements

This study was supported by Samsung Display Corporation, the Centre for Advanced Soft-Electronics (Grant 2013M3A6A5073183), Ministry of Science and ICT through the National Research Foundation, grant-funded by the Korea government (2017R1E1A1A01075360, 2020R1A2C4001617, and 2020R1A4A1019455).

## Author contributions

A.L., M.-G.K., and Y.-Y.N. designed the research and experiments. A.L. and H.H.Z. fabricated the devices. A.L., H.H.Z., and W.-T.P. performed the characterisation and analysis. S.-J.K. and H.K. performed the DFT calculation. A.L., H.H.Z., M.-G.K., and Y.-Y.N. wrote the paper. All authors contributed to the discussions. A.L. and H.H.Z. contributed equally to this study.

## Competing interests

The authors declare no competing interests.
