## [Peer Review File · Nature Communications]

REVIEWER COMMENTS

Reviewer #2 (Remarks to the Author):

The authors have presented us an easy way to achieve high mobility p-type semiconductor for thin-film transistor applications. We agree with the authors that high performance p-channel TFTs are very essential for various applications. We support publication of this paper as long as the authors could fix some minor issues.

1. In terms of illumination decomposition, the authors chose 550 nm (~ 2.3 eV) as well as UV light sources and saw distinct results. I am interested in the stability of CuI under blue light (~ 3 eV), which corresponds to most commonly used blue-light LEDs.
2. Generally speaking, amorphous films are more favorable than polycrystalline ones for TFT applications despite their higher carrier mobility. For instance, applications of LTPS are restricted to small area displays despite its high mobility (> 100 cm²V⁻¹s⁻¹). When the channel length is small enough, inhomogeneity of different devices becomes intolerable. Can the authors comment on this issue? Does polycrystalline CuI have the same problem?
3. Please give us more details about how IGZO TFTs are bonded to CuI TFTs with probe.
4. The authors mentioned that photolithography process would degrade the quality of CuI thin film. So how did the authors deposit electrodes? Does this involve photolithography?
5. The authors have shown some merits of the inverter circuits with a large V_{dd}, which is not practical in real applications. At this V_{dd}, static power consumption is just too large. We would like to see more details at smaller V_{dd} (10V, 30V) in Fig. 4b.
6. In Fig. S22, it seems that device performance improved greatly after passivated by CYTOP. That is quite encouraging. Please show us some details about this passivated device, i.e. carrier mobility, threshold voltage.

Reviewer #3 (Remarks to the Author):

CuI is a current hot topic in semiconductor research because it is an intrinsic p-type conducting material, which are rare among the compound semiconductors. In addition, CuI is optically transparent. The precise control of conductivity is a challenge with CuI, therefore the concept of the authors to dope with Zn is highly interesting and innovative. Furthermore, the fabrication of field effect transistors (FETs) is challenging too, and therefore I highly support publication of this manuscript, even if not all parameters of the FETs are optimized for all kind of applications, such as for example the currently required high gate voltages. The authors claim an interesting low-temperature solution-based doping approach for Zn cations as hole suppressors for CuI.

To my opinion, this manuscript has the following highlights:

1. Inclusion of theoretical DFT calculations for control of Fermi level for various dopants of CuI
2. Zn dopant established as the superior one related to several other dopants, see lines 111, 112. Linear Zn implementation up to 15 mol% (Fig.2c)
3. weak lattice distortion and no impurity phases up to 5 mol% Zn in CuI, relatively smooth film surface.
4. successful FET preparation with high hole mobility and high on-off-ratio, Fig.3.
5. precise film thickness control in the 4-11 nm range for precise control of FET characteristics Fig. S11.
6. Very good wafer scale uniformity of FET characteristics (Fig. 4 a, b, c, S24)
7. Integration into a logic inverter together with a a-IGZO FET with a high gain of 56 (Fig. 4d, e, f).
8. very extensive supplementary material with lots of very useful detailed results - this is really good.
9. effect of a top capping layer for better long-term stability over few days is well demonstrated (Figs. S20, S21, S22)

All these highlights are technologically the state of the art in this special research subject.

Comments for consideration by the authors:

In lines 156-158 the authors claim a homogeneous Zn distribution over the CuI film thickness. However, the EDS images and the SIMS profile show some weaker inhomogeneities of the Zn distribution with a probably

weak enrichment at the interface to SiO₂. So the above claim (lines 156-158) should be expressed little bit weaker.

Furthermore, in Figs. 2f, S8, and S9 should be made clear either in the Figures or in the captions where (which side) is the substrate (SiO₂/Si) of the CuI:Zn film. This is not clear now.

Because the manuscript in its current form is already the second revised version of this work, I am otherwise happy with the concise presentation in good language and recommend publication after minor revision.

Reviewer #4 (Remarks to the Author):

Liu et al. show a low-temperature deposited semiconductor which is proposed to act a p-type counterpart to amorphous IGZO in flexible and transparent electronic devices. The presented characterization of the CuI semiconductor film is very extensive and shows good results. The shown single TFTs at the same time are very basic, they are good enough to demonstrate the functionality but a more detailed electrical characterization (e.g. gate leakage current, linear & output lines for all measurements, gate capacitance) would be needed to judge their usefulness in systems. It would be an advantage if the shown devices would actually be transparent or flexible.

Additional, more technical comments are as follows:

- From fig. 2f it is not clear where the semiconductor layer is exactly located, there seems to be Zn everywhere (with two parallel bands of slightly higher concentration). This should be discussed.
- The mobility is partially confusing, there is a field effect mobility, a saturation mobility and a linear mobility. First, what is the used definition of field effect mobility? Second, what is the reason for the discrepancy between linear and saturation mobility? Third, Hall measurements are mentioned, what is the measured Hall mobility?
- The manuscript gives the impression that all semiconductor islands were structured by manually scratching the CuI layer with a needle because other methods damage the layer. Is this also true for the shown arrays? What is the yield for this process? The implication concerning large area fabrication should be discussed. A micrograph of such a device would be helpful.
- The shown TFTs are depletion mode devices, can you discuss the possibility to realize more useful (in particular for digital circuits) enhancement mode TFTs?
- It is good that hysteresis measurements are shown but are the parameters extracted for the forward or reverse sweep? Would there be a difference as the hysteresis is relatively small?
- It was written that a CMOS inverter was integrated. How is this possible if the CuI layer cannot be structured? Can you include a picture of the circuit?

Reviewer #1 (Remarks to the Author):

The authors describe in their paper the strong impact of refractive index change by absorption heating on the scattering intensity of nanoscale Mie-resonators. Depending on the size of the resonators a substantial positive or negative deviation from the usual linear behaviour is observed which was clearly observed in dark field microscopic measurements. Evaluating the local temperature of the Mie-resonators from Stokes-Antistokes Raman spectra the authors could attribute this to the temperature dependent refractive index change of the Si. Taking also the temperature dependent absorption cross section of the Mie-resonators into account a simulation of the temperature increase in the Mie-resonators was carried out. The modelled scattering of the Mie-resonators based on this temperature affected refractive index then lead to a very good correspondence with the experimental results. Finally the fast switching of the nonlinearity could be demonstrated with time resolved pump probe measurements and a narrowing of the point spread function due to the linearity was also presented.

The experimental results and theoretical simulations are convincing and present a coherent picture of the effect. The fast relaxation of the nonlinearity in the ns- range due to the small overall thermal capacity (fast cool down) is surprising and demonstrates impressively that thermal switching can be fast, when it is applied to nanoscale volumina. In addition the results show, that high Q-resonances (e.g. from photonic crystals defects) are not necessary to employ nonlinearities and that small simple Mie-resonators can already be of use. With the current interest in Mie-resonator based dielectric metamaterials the topic of the paper appears timely. The added information in the supplementary material supports the claims of the paper further and demonstrates that the authors investigated the presented effects thoroughly and extensively.

Overall the paper reads well and should be accessible to a large community of readers. However there are still a few shortcomings which the authors must address:

1) The critical point for the understanding that there is for some particles a negative nonlinear deviation of scattering intensity (100nm, 190nm) and for the 170nm a positive nonlinear deviation of scattering intensity is not given in the paper. In the moment it only appears in the supplementary material as caption on Fig E3: "Since Mie scattering spectrum red-shifts at elevated temperature, this explains...". This crucial explanation has to be also well presented in the main paper! Only then the reader can understand why the nonlinearity acts sometimes as increasing and sometimes as decreasing. The spectral shape of the scattering spectra with the different resonances is crucial here.

We appreciate the reviewer's positive comments and we agree with the reviewer that spectral information of the nanoparticles at different temperatures should be given to enhance the clarity. In the revised figure 3e, 3f, and 3g, we have added temperature-dependent spectral shifts to explain the physical origin of the negative/positive nonlinear deviations.

Fig. 3 e-g...The inset of each figure presents the corresponding resonance spectrum shift at elevated temperatures.

(p. 9, third paragraph)

The negative nonlinear deviation (100-nm, 190-nm) and positive nonlinear deviation (170-nm) are well explained by the temperature-dependent resonance spectrum shift in the insets.

2) What is the influence of the free carriers on the refractive index. In the paper only the thermal impact on the refractive index is considered. However as the applied light wavelength (561nm) is well absorbed by the Si, a lot of free carriers are generated and should also have an impact on the refractive index or can this be neglected? The authors should estimate the impact of the free carrier concentration on the refractive index and describe the outcome of it in the supplementary material.

This is a good point. In pulsed laser excitation scenario (Fig. 4b), we have explicitly shown that two relaxations in time were observed. Free carriers are first excited by pulsed lasers, and they lead to index changes. Subsequently, the temperature rises due to the carrier-phonon scatterings and recombination of free carriers. The fast relaxation (10s of picoseconds) corresponds to free carriers and the subsequently slow relaxation (nanosecond) represents thermal effect. That is, free carriers do have an impact on refractive index, as the reviewer suggested, but the effect can be clearly distinguished from thermal response in time domain.

Furthermore, in Fig. 2 and 3 of the manuscript, we used CW lasers, whose intensity is 4-5 orders lower than the peak intensity of mode-locked laser excitation in Fig. 4b. Since the density of free carrier should be proportional to excitation intensity, in CW excitation case, free carrier effect should be negligible. This is verified by the outstanding agreement between experiment (Fig. 2a-2c) and photothermal theory (Fig. 3e-3g).

(p. 12, first paragraph)

... The response of the free carrier and photothermal effect is clearly distinguished in time domain. Note that in Fig. 2 and Fig. 3, where CW lasers are applied, the laser intensity is 4-5 orders lower than that of Fig. 4b, and thus, the free carrier effect is negligible....

3) Fig 4c and explanation of resolution enhancement by SAX in the supplementary materials: From the the explanation in the supplementary materials it is not clear how the resolution enhancement is exactly measured and how the curve in Fig. 4c of the main paper is actually

obtained. It is mentioned that the two beams from the AOMs are interfered and “generate a sinusoidally modulated illumination beam at 10kHz frequency”. Does this mean that a spatially modulated light pattern is generated whose intensity is additionally oscillating with a frequency of 10kHz in the time domain? The authors should explain in more detail how the enhanced resolution (reduced PSF) is actually measured since from the existing explanations this appears to be a quite complex procedure. Maybe a schematic drawing of the setup, which was used for the SAX-measurements will help.

We apologize for not explaining the setup clearly. The modulation is only in temporal domain, not in spatial domain, and the purpose of using two AOMs is simply to generate a pure sinusoidal modulation [see ref. 31 of the main text]. Following the reviewer’s suggestion, we have added a setup scheme in supplementary Fig. 13, to explain in detail how to achieve the enhanced resolution.

If the authors can improve/clarify these mentioned points, I recommend the paper for publication.

Reviewer #2 (Remarks to the Author):

In the paper "Giant optical nonlinearity in single silicon nanostructure: ultrasmall all-optical switch and super-resolution imaging" authors show the all-optical tuning of the Si-blocks that support the excitation of Mie resonances. The paper is well written and explains everything in a detailed manner. It shows comprehensive studies on the photothermal nonlinearity in silicon nanoblocks, that originate from the Mie resonance excitation. Authors provide a large amount of data including nonlinear measurements for different dimensions of silicon blocks, Raman measurement, ellipsometry measurements, and others. Authors utilize Mie resonances to enhance the absorption of Silicon and to achieve 5-orders of magnitude enhancement of nonlinearity. Paper looks interesting for the metamaterial community, however, I am concerned about novelty and if it is suitable for Nature communications, as well as believe this paper requires some revisions.

1) As for the novelty please read recently published paper, Berzinš, Jonas, et al. "Laser-induced spatially-selective tailoring of high-index dielectric metasurfaces." *Optics Express* 28.2 (2020): 1539-1553.

Apart from the spectral tuning of the Mie resonances due to the heating, it also provides additional data about the melting/reshaping and damage threshold of Silicon metasurface. Please, emphasize the differences in your results.

We appreciate the reviewer to compare our work with recently published papers. The main difference is that the OE paper describes a non-reversible effect, where the silicon nanoparticle shape and the corresponding Mie resonance are permanently changed by picosecond laser induced heat. In contrast, our photothermal nonlinear effects do not involve any nanostructure shape change, and are fully reversible.

(p. 6, last paragraph)

The reversibility and repeatability of the nonlinear responses are demonstrated in Fig. 2e and Fig. E6, thus excluding the possibility of non-reversible oxidation or shape change of silicon²⁴.

2) Moreover, another paper has relevant studies that you do not cite. It provides the result of the interplay between free-carrier refractive index change, and thermal heating in Si-metasurface: Della Valle, Giuseppe, et al. "Nonlinear anisotropic dielectric metasurfaces for ultrafast nanophotonics." *ACS Photonics* 4.9 (2017): 2129-2136.

We thank the reviewer to mention this reference, that we are aware of. Although this reference discussed photothermal effect of silicon nanoparticles, it aimed to minimize photothermal effect at selective wavelength to achieve high-speed modulation based on free carriers. In this reference, the modulation speed reaches picosecond scale, but modulation depth is less than 10% (by free carrier), at a much higher excitation intensity (>GW/cm², or equivalently >10W/um²). Their reported photothermally induced modulation is always less than 1%. In contrast, our work demonstrates >400% variation of scattering with a CW laser at only few mW/um²! That is, their nonlinear responses are much smaller than ours.

(p. 11, last paragraph)

Note that our reported photothermal nonlinear response in an isolated low-Q silicon resonator is much higher than that in a thermally coupled nanostructures on a metasurface **with free-carrier effect**³³ or with Fano resonance (Q-factor ~1000).¹⁹

3) And another paper that can be cited: Bosch, M., et al. "Polarization states synthesizer based on a thermo-optic dielectric metasurface." *Journal of Applied Physics* 126.7 (2019): 073102.

We have cited this article, along with several other thermo-optic papers, as a potential application direction in the future.

(p. 12, 3rd paragraph)

In addition to the above applications, recently, there has been an emerging trend using thermo-optic effect to modulate metamaterial optical properties, such as polarization³⁵, miniaturized image contrast³⁶, and metalens focal length³⁷. We envision that our result manifests the potential of pushing these various modulations toward all-optical control.

Other questions:

1) The authors provide the main experimental results in the numbers of NDR – nonlinear deviation. I can see how it can have a mathematical meaning, but from the experimental point of view, this result can lead readers to confusion. For example, line 98, authors claim that you have “400% enhancement or 70% reduction of scattering, i.e.”. Yes, it is defined that “i.e. deviation”, but, it’s not a 400%/70% change of scattering. I can accept NDR after I read what does it mean; however, if I think carefully, you calculate deviation from some number that cannot be achieved in the experiment. Moreover, the most important result is shown in Fig.3 – change of scattering efficiency of the nanoblock under different excitation power. Here we can clearly see that scattering changes and NRD has nothing in common, – the main changes of scattering occur at 190 nm width of the block. But NDR from 190 nm block has the smallest value. Therefore, please provide some other papers that use NDR value, since I admit, that I might not be familiar enough with that area, or please, consider emphasizing other experimental values.

We agree that no other literature use NDR, but we have a good reason for it. Before I explain our reason, I hope to clarify the meaning behind the reviewer’s statement: “(in Fig. 3) the main changes of scattering occur at 190 nm width of the block. But NDR from 190 nm block has the smallest value” Please note that the meaning of NDR is deviation from linear extrapolation, and it would be zero when there is no deviation. In Fig. 3, NDR from 190 nm block has the value of -70%, which represents a large deviation, in good agreement with the scattering change pattern in Fig. 3g. That is, NDR serves as a reasonable indicator of the nonlinearity that includes both amplitude and sign.

Conventionally, in the field of nonlinear optics, pulse lasers were adopted, and typically the pulse lasers are split into pump and probe beams. With pump-probe experiments, nonlinear response is defined as $\Delta I/I_0$, where I_0 is the signal intensity without pump pulse and ΔI is

that with pump pulse. In the case of measuring scattering, the pump-probe nonlinear response is defined as $\Delta S/S$.

Now in our case, we used a CW laser for efficient heating, and the same laser beam provides nonlinear characterization. Therefore, it is not necessary to split into pump and probe beams, and thus the definition of our nonlinear response would be different from most existing literatures. Having said this, as we explained in Fig. 2a, NDR is in fact the same as $\Delta S/S$, by defining S as linearly extrapolated value, and ΔS is deviation from S. To further emphasize this similarity, we have added the notion that $NDR = \Delta S/S$ in all relevant figures (Fig. 2a, 2b, 2c, 2f; Fig. 3d, 3e-g; Fig. 4b).

2) Moreover, when you provide a comparison with other works, you use NDR. None of these papers in refs [19,21] use this value to describe the modulation of the signal. There are two values that majority of community uses: it is the absolute value modulation of optical signal (ΔI), or the relative modulation of the signal, $\Delta I/I_0$, where $\Delta I = I_{\text{pump}} - I_0$, I_0 is the unperturbed reflectance/transmittance and I_{pump} is the reflectance/transmittance when high-intensity laser is used – mostly for better visualization of the changes (Please see papers from references [19,21], or another paper that also shows I-scan, and authors use absolute values: Zubyyuk, Varvara V., et al. "Low-power absorption saturation in semiconductor metasurfaces." ACS Photonics 6.11 (2019): 2797-2806.) Therefore, please correct the comparison of the results with the previous works.

As we mentioned in the last response, our definition of NDR is the same to $\Delta I/I_0$ in the previous references (I_0 is linearly extrapolated in our case). Therefore, we compare directly our NDR with the value of $\Delta I/I_0$ in other references.

3) Since you use CW laser, and other papers use pulsed laser, please, provide how you recalculated their excitation intensity, for example, in supplementary materials.

We have included the recalculation in the Methods section, under the title of "Transient measurements by ultrafast techniques" in page 16 of the main text, which is based on the following equation:

$$\begin{aligned} \text{Intensity} &= \frac{\text{pulse energy}}{(\text{pulse width})(\text{focus area})} \\ &= \frac{\text{laser power}}{(\text{repetition rate})(\text{pulse width})(\text{focus area})} \end{aligned}$$

For example, in Fig. 4 of Ref. 21, they achieved <1% transmission modulation by using pump fluence of 30 $\mu\text{J}/\text{cm}^2$, and 45 fs pulse width. The corresponding excitation intensity is $30/45 * 10^9 \text{ W}/\text{cm}^2 \sim 6.7 * 10^4 \text{ mW}/\mu\text{m}^2$, which is much larger than our excitation intensity in CW case.

4) I believe that the non-gaussian form of your results can come from the spectral tuning of the resonance. If you change the refractive index, the spectral position of the Mie resonances will tune, as I mentioned in the paper Berzinš, Jonas, et al. "Laser-induced spatially-selective tailoring of high-index dielectric metasurfaces." *Optics Express* 28.2 (2020): 1539-1553, or any other refractive index change papers. It will be relevant to show numerical calculation of spectrum at least for one sample, to show how its spectrum is going to be modified when you change the refractive index of silicon due to the photothermal effect. Therefore, it will give a better explanation of the achieved results.

We agree with the reviewer, and added simulated spectrum tuning in the revised Fig. 3e-3g, to provide a better explanation of the nonlinearity.

5) Please, provide more details regarding the nanoblock sample used:

- How far are the nanoblocks from each other?
- Is there any coupling between them that can modify the spectra of Mie resonances?

The distance between each nanoblock is 5 micrometers, and thus the coupling among them is negligible. The information is added in p. 3, last paragraph.

6) Other small corrections:

When you provide the number of nonlinearities, for example on line 64/66, please provide references.

We have added references for Kerr nonlinearity (ref 3) and photothermal nonlinearity (ref 5) of bulk silicon.

- Leuthold, J., Koos, C. & Freude, W. Nonlinear silicon photonics. *Nat. Photonics* **4**, 535–544 (2010).
- Horvath, C., Bachman, D., Indoe, R. & Van, V. Photothermal nonlinearity and optical bistability in a graphene–silicon waveguide resonator. *Opt. Lett.* **38**, 5036 (2013).

I believe lines 88-90 are in general incorrect. In this review you separate metasurfaces from nanoresonators, but if I am not mistaken, in all this works authors use metasurfaces as arrays of nanoresonators.

We thank the reviewer for careful reading. The sentence is modified as " For example, Si metasurfaces can be applied for fabrication of ultracompact phase controllers,¹⁹

several order-of-magnitude enhancement in third harmonic generation,^{20,21} and two-photon absorption.²²

Also, I recommend considering changing the title of the paper, since there is no “photothermal” anywhere.

Agree, we modify the title to be "Giant **photothermal** nonlinearity in a single silicon nanostructure", as suggested by the third reviewer

Altogether, the work is of interest to the metamaterial/silicon community and can be published after revision.

Reviewer #3 (Remarks to the Author):

In their manuscript, Duh et al. report on the photothermal response of Mie-resonant silicon nanoparticles. I think this is a technically sound study revealing an interesting property of a well-studied system: light can efficiently 'tune' itself in an ultra-small particle by heating and exploiting the thermo-optic response of silicon. The authors proceed to demonstrate several intriguing applications of this effect for switching the scattering efficiency and reducing the spot size of a focused laser beam down to 132 nm at a wavelength of 592 nm. I think that the presentation quality, the rigor of the research effort, as well as the timeliness of the topic can secure publication in Nature Communications after the following issues are addressed:

1. The title of the work needs to be more specific about the nature of the nonlinearity. I suggest using the following wording: "Photothermal nonlinearity." Also, there is no need to emphasize the fact that it is an ultrasmall all-optical switch: the fact that a small silicon nanoparticle may show all-optical switching was discussed at length in Refs [19, 21, 26]. Finally, I am not convinced that the paper shows any explicit 'super-resolution imaging;' a mere tight focal spot demonstration in Fig. 4 is not enough to claim super-resolution imaging, especially in the title. Therefore, I recommend changing the title of the paper to "Giant photothermal nonlinearity in a single silicon nanoparticle." This way is it significantly more concise, and the main novelty is properly emphasized.

Agree, we modify the title to be "Giant photothermal nonlinearity in a single silicon nanostructure", as suggested by the third reviewer

2. The way the silicon films were obtained is unclear. The authors used "a 150-nm-thick monocrystalline Si layer on the quartz substrate." Having a monocrystalline Si film on a quartz substrate is unusual. I could not locate this product on the website of the supplier the authors acknowledged. Therefore, I suggest adding to the Methods section of the paper the procedure of how these silicon films were deposited, for the sake of study reproducibility.

We have added a reference and description in the Methods section.
(p.12, last paragraph)

The 150-nm-thick monocrystalline silicon on quartz substrate (Shin-Etsu Chemical Co., Ltd.) was fabricated by wafer bonding at the temperature of $T < 1000$ degree after H⁺ ion implantation to Si wafer surface.³⁸

3. Please use caption to annotate the variables in Fig.1, such as CscF and Cabs.

The corresponding annotation is added in the caption of Fig. 1.

e. Simulated single-particle forward scattering (CscF) and f. absorption cross section (Cabs) spectra versus different nanoblock widths.

4. Several stylistic suggestions would include getting rid of informal expressions, such as 'aka' (line 38), 'etc' (line 59), as well as unnecessary neologisms such as 'meta-silicon-material' (line 81), 'nano-silicon applications' (line 103) and others. In general, I find that the English of the paper is rough, and I strongly recommend invoking an English-editing service.

The manuscript was English edited as requested.

REVIEWERS' COMMENTS:

Reviewer #2 (Remarks to the Author):

The authors have addressed my concerns which I raised previously. I still have some concerns about the long-term stability of devices with under blue light. However, given the high field effect mobility and the industrialization potential of Zn-CuI system, as well as the extended device stability with proper encapsulation. I believe the work is of great interest in the field of thin-film transistors and I support publication of this work in Nature Communications.

Reviewer #3 (Remarks to the Author):

To my opinion, the authors revised the manuscript satisfactory, and I recommend now to publish it as it is. An excellent and helpful highlight is the extensive supplementary information with 26 Figures. The subject of CuI-based transistors is a current hot topic and interesting to the semiconductor and solid state physics community. This work demonstrates some valuable progress in fabricating such devices.

Reviewer #4 (Remarks to the Author):

Liu et al. gave satisfying answers to basically all reviewer comments. At the same time, these are not always reflected in the manuscript or supplementary information (e.g. various figures from the rebuttal letter). I would avoid the impression that less positive aspects of a study are actively hidden. I realized the following:

1. The scratching fabrication approach (which is basically not scalable) is not clear from the manuscript and should be properly explained.
2. Based on the reply to Q1 of reviewer 4 it sounds like Fig. 2f shows noise or some kind of artifact. So why is it included in the manuscript?
3. Explanation of the exact inverter structure. There is a big difference between an integrated CMOS circuit and two TFTs connected by cables. Here, the circuit is not integrated but the opposite is written in the paper.
4. If the different mobility values are due to aging and/or instability it should be mentioned, just having a graph with different values (Fig. 3f) is very confusing.

Comments of Reviewer #2

The authors have addressed my concerns which I raised previously. I still have some concerns about the long-term stability of devices with under blue light. However, given the high field effect mobility and the industrialization potential of Zn-CuI system, as well as the extended device stability with proper encapsulation. I believe the work is of great interest in the field of thin-film transistors and I support publication of this work in Nature Communications.

Comments of Reviewer #3

To my opinion, the authors revised the manuscript satisfactory, and I recommend now to publish it as it is. An excellent and helpfull highlight is the extensive supplementary information with 26 Figures.

The subject of CuI-based transistors is a current hot topic and interesting to the semiconductor and solid state physics community.

This work demonstrates some valuable progress in fabricating such devices.

Comments of Reviewer #4

Liu et al. gave satisfying answers to basically all reviewer comments. At the same time, these are not always reflected in the manuscript or supplementary information (e.g. various figures from the rebuttal letter). I would avoid the impression that less positive aspects of a study are actively hidden. I realized the following:

Reply: Firstly, we really appreciate the reviewer's kind revision and helpful comments in the past few rounds of revision. Based on the reviewer's suggestions, more detailed process/structure descriptions were added accordingly.

Q1. The scratching fabrication approach (which is basically not scalable) is not clear from the manuscript and should be properly explained.

Reply: Thanks for the reviewer's reminding. The more detailed description about the scratching process was added in the manuscript. The visual schematic diagram and patterned channel image were added in Figure S12 (also shown below).

Page 8→The channel layers were patterned through probe scratching to guarantee low gate leakage current.

Q2. Based on the reply to Q1 of reviewer 4 it sounds like Fig. 2f shows noise or some kind of artifact. So why is it included in the manuscript?

Reply: Thanks for the reviewer's comments. In Fig. 2f, due to the small amount of Zn^{2+} doping in CuI, the Zn element tracking using TEM mapping may be influenced by the background noise. Therefore, we further measured sample SIMS data (Fig. S10, also shown below) to confirm the uniform distribution of Zn^{2+} in CuI.

According to reviewer's suggestion, we removed the TEM mapping images to the Supporting information (Figure S9) and reorganized the Fig. 2f with only clear TEM images (as shown below).

Q3. Explanation of the exact inverter structure. There is a big difference between an integrated CMOS circuit and two TFTs connected by cables. Here, the circuit is not integrated but the opposite is written in the paper.

Reply: Thanks for the reviewer's suggestion. We corrected the statement using 'assembled' instead of 'integrated'. In addition, more detailed description about the inverter connection was added in the revised manuscript. The diagram description of inverter connection using cable was also added in Figure S25b.

Page 12→a proof-of-concept complementary inverter was assembled by cables to analyse the potentials of CuI:Zn TFTs for the realisation of high-performance logic circuits.

Q4. If the different mobility values are due to aging and/or instability it should be mentioned, just having a graph with different values (Fig. 3f) is very confusing.

Reply: Thanks for the reviewer's comment. In the last response letter, we speculated one possible reason for different mobility values of low-temperature measurement, because this measurement was conducted in another lab, the samples may contact with the air during transfer process. After the reviewer pointed out the mobility discrepancy, we carefully encapsulated the TFTs and measured again and got the precise results. The mobility values given in Fig. 3f were extracted from the transfer curves shown in Figure S23.